# Elastic Taping Application on the Neck: Immediate and Short-Term Impacts on Pain and Mobility of Cervical Spine

**DOI:** 10.3390/jfmk8040156

**Published:** 2023-11-07

**Authors:** Luca Russo, Tommaso Panessa, Paolo Bartolucci, Andrea Raggi, Gian Mario Migliaccio, Alin Larion, Johnny Padulo

**Affiliations:** 1Department of Human Sciences, Università Telematica Degli Studi IUL, 50122 Florence, Italy; l.russo@iuline.it (L.R.); p.bartolucci@iuline.it (P.B.); 2Department of Biotechnological and Applied Clinical Sciences, University Degli Studi dell’Aquila, 67100 L’Aquila, Italy; tommaso_panessa@libero.it; 3Laboratory of Biomechanics, FGP srl, 37062 Verona, Italy; andrea@fgpsrl.it; 4Department of Human Science and Promotion of Quality of Life, San Raffaele University, 00166 Rome, Italy; 5Faculty of Physical Education and Sport, Ovidius University of Constanta, 900029 Constanta, Romania; alinlarion@yahoo.com; 6Department of Biomedical Sciences for Health, Università degli Studi di Milano, 20133 Milan, Italy; johnny.padulo@unimi.it

**Keywords:** cervical ROM, elastic taping, neck pain, kinesiology, musculoskeletal health

## Abstract

The aim of this study was to measure the effects on three-planar active cervical range of motion (ACROM) and self-perceived pain of elastic taping (ET) application in the cervical area. Thirty participants (*n:* 22-M and 8-F, age 35.4 ± 4.4 years; body height 173.1 ± 8.4 cm; body mass 73.5 ± 12.8 kg) in the study group (SG) and twenty participants (*n:* 11-M and 9-F, age 32.6 ± 3.9 years; body height 174.9 ± 10.9 cm; body mass 71.2 ± 12.9 kg) in the control group (CG) were recruited. All subjects had neck and cervical pain in baseline condition. Each group performed an ACROM test and measured the perceived pain in the neck based on the Numerical Rating Scale (NRS 0--10, a.u.) at the baseline (T0), after 20′ from the ET application (T1), and after three days of wearing the ET application (T2). Between T0 and T1, an ET was applied to the cervical area of the SG participants. Statistical analysis did not show any significant change in CG in any measurement session for ACROM and neck pain parameters. Conversely, the SG showed significant improvements for ACROM rotation to the left (T0 64.8 ± 7.7°–T2 76.0 ± 11.1° *p* < 0.000) and right (T0 66.0 ± 11.9°–T2 74.2 ± 9.6° *p* < 0.000), lateral inclination to the left (T0 37.5 ± 6.9°–T2 40.6 ± 10.8° *p* < 0.000) and right (T0 36.5 ± 7.9°–T2 40.9 ± 5.2° *p* < 0.000), extension (T0 47.0 ± 12.9°–T2 55.1 ± 12.3° *p* < 0.001), and flexion (T0 55.0 ± 3.6°–T2 62.9 ± 12.0° *p* < 0.006). A significant decrease was also measured in SG for pain NRS between T0 and T2 (T0 7.5 ± 1.0°–T1 5.5 ± 1.4–T2 1.4 ± 1.5° *p* < 0.000). In conclusion, a bilateral and symmetrical ET cervical application is useful to enhance multiplanar ACROM and reduce subjective self-perceived cervical pain when it is needed. Based on the evidence, the use of ET on the neck is recommended for managing neck motion restrictions and pain in adult individuals.

## 1. Introduction

The cervical spine, consisting of seven vertebrae, is known for its exceptional mobility within the spinal column. It plays a crucial role in facilitating multiplanar movements of the head in space. Specifically, the motion of the cervical spine and head complex can be categorized into right–left rotation, right–left lateral inclination, flexion, and extension, meaning forward and backward head motion, respectively [1]. These movements can be performed individually on a single anatomical plane or via combined multiplanar actions, depending on the functional demands of the environment.

Given these unique characteristics, preserving or restoring the physiological active range of motion (ACROM) of the cervical spine becomes a significant challenge in daily life, particularly in sports and various activities, as it contributes to maintaining optimal health [2,3]. Numerous factors can impede and restrict the ACROM, including traumatic injuries, repetitive movements, occupational practices, prolonged postures, head positioning, and other factors [4,5,6,7,8]. Diminished ACROM can have various detrimental effects on the musculoskeletal system, affecting the use of the eyes, other spinal segments, shoulder mobility, etc. Additionally, a reduced ACROM is a common observation among individuals experiencing neck pain [9,10,11,12,13]. In light of this context, two primary considerations arise: (1) the need for a comprehensive evaluation of the ACROM and (2) the implementation of effective strategies to enhance it.

Regarding the ACROM assessment, professionals have the option to conduct tests with or without devices [14]. Qualitative analyses without devices are cost effective and straightforward, but there is no change to compare data over time [15]. Within the realm of instrument-based evaluation, various solutions exist, including video analysis and 3D motion capture systems. However, these methods are more suited to analyzing posture and complex movements in specific contexts [16,17,18]. As a result, the most utilized and practical approach involves the use of inertial sensors [3,19,20,21,22].

These tools enable professionals to accurately measure the ACROM in a convenient and efficient manner, without causing discomfort to the subjects. Typically, an inertial sensor is positioned on the subject’s forehead and moves in conjunction with the head, capturing angles and the range of motion during neck active movements [19,20,21,22], avoiding any compensative movement of the shoulder girdle.

Regarding strategies for preserving and improving ACROM, multiple approaches can be considered, such as reducing screen time, minimizing sedentary behavior, receiving massages, engaging in targeted exercises, utilizing cupping techniques, and employing elastic taping [23,24,25,26,27]. Among these solutions, elastic taping (ET) possesses a distinctive characteristic: it can be worn directly on the skin for an extended period, continuously working 24 h a day and adapting to the subject’s movements until its removal. This unique characteristic makes ET a valuable and practical tool not only for sportspeople and athletes [28,29] but also for individuals in their daily lives.

Based on the current knowledge, the effectiveness of ET applications in reducing neck pain and improving the ACROM has been demonstrated [30,31,32]. However, there is a lack of data on the immediate and short-term effects of a single bilateral and symmetrical ET application worn for three days by video terminal workers experiencing neck pain and restricted ACROM. The hypothesis of this research is that wearing the same ET application for three days can have beneficial effects on cervical pain and ACROM outcomes in video terminal workers. Therefore, the objective of this research was to assess the immediate and short-term impacts of a single bilateral and symmetrical ET application on self-perceived pain and multiplanar motion of the cervical spine.

## 2. Materials and Methods

### 2.1. Design and Participants

This study utilized a short-term longitudinal small-cohort design with repeated measures. The participants were recruited from employees of a call center located in the south of Italy. Voluntary participation was sought, and specific selection criteria were applied, including (1) individuals experiencing recurrent self-perceived musculoskeletal cervical pain (recurrence more than 1 episode per year), (2) individuals receiving no treatments for cervical pain before or during the research period, and (3) individuals working as video terminal operators.

A total of 60 participants were recruited and randomly assigned to either the study group (SG) or the control group (CG), ensuring a balanced distribution. However, 10 participants dropped out during the course of this study, resulting in a final sample size of 50 participants completing the experiment (Figure 1). The SG comprised 30 participants (*n:* 22-M, age 35.6 ± 4.4 years; body height 176.6 ± 6.6 cm; body mass 79.3 ± 9.1 kg and 8-F, age 34.8 ± 4.5 years; body height 163.5 ± 4.3 cm; body mass 57.6 ± 6.0 kg). The CG included 20 participants (*n:* 11-M, age 32.8 ± 3.6 years; body height 182.3 ± 7.5 cm; body mass 79.0 ± 9.2 kg and 9-F, 32.2 ± 4.5 years; body height 165.9 ± 6.8 cm; body mass 61.7 ± 10.4 kg). A power analysis was performed, indicating that sample sizes of 20 and 30 subjects per group, respectively, would provide 80% power, with a 5% error probability and an effect size of 0.55. Prior to the intervention phase, all participants were thoroughly informed of this study’s purpose and provided voluntary consent. Privacy criteria were also strictly upheld. This study received approval from the Ovidius University of Constanta, Number 78, on 27 January 2023, in accordance with the principles outlined in the Helsinki Declaration.

### 2.2. Instrumentation

The assessment of multiplanar ACROM was conducted using an inertial sensor (Moover^®^, Sensor Medica, Guidonia-RM, Italy) positioned in the middle of the forehead and secured with an elastic band (Figure 2).

To ensure the accuracy and validity of the inertial sensor, a preliminary comparison was made using a six three-dimensional camera optoelectronic system (SMART DX, BTS Bioengineering, Garbagnate Milanese, Italy), which served as the gold standard. A convenience sample of nineteen subjects participated in this trial, performing ACROM tests using the inertial sensor with a passive reflective marker attached to it. The data obtained from the Moover^®^ sensor did not show any statistically significant differences compared to the 3D kinematic data. Further details can be found in Appendix A.

Self-perceived cervical pain was evaluated using the Numerical Rating Scale (NRS, 0–10), with 0 representing the absence of pain and 10 indicating the highest level of sustainable pain.

For the ET application, two personalized strips of Taping Elastico^®^ (ATS, Arezzo, Italy) were used for each subject in the SG. The application procedures are described in the next section.

### 2.3. Procedure and Data Collection

The testing procedures were conducted in a dedicated room within the participants’ working place, maintaining a mean temperature of 19 °C and a mean relative humidity of 52%. To minimize the potential influence of circadian effects, each subject underwent testing at the same time of day, as is customary in laboratory procedures of this nature [33]. This study consisted of three test sessions (T0–T1–T2), with a preliminary familiarization session conducted one week prior to the start of the protocol to provide instructions to the participants. T0 served as the baseline assessment, T1 served as the acute assessment taken 20 min after the application of the ET on the cervical area, and T2 served as the short-term assessment conducted after three days of wearing the ET application (Figure 3). T0 and T1 assessments were performed on the same day. Both SG and CG underwent the three test sessions in the same order. The CG received no intervention between T0 and T1, as well as between T1 and T2.

Each test session (T0–T1–T2) consisted of two evaluations performed in the following sequence: (1) the assessment of perceived cervical pain using a 0–10 Numerical Rating Scale (NRS) and (2) the measurement of ACROM using an inertial sensor.

The NRS was printed on white paper, and each subject marked an “X” on the score corresponding to their self-perceived pain in the cervical area.

The assessment of multiplanar ACROM involved measuring angular motions in three directions: right–left rotation on the transversal plane, right–left lateral inclination on the frontal plane, and flexion–extension on the sagittal plane (Figure 4). Each test included a total of 14 movements, with 7 repetitions performed for each direction. The maximum and minimum values were excluded, and the average value was calculated. During the tests, each subject was seated to stabilize the hips and the lumbar spine, while the entire trunk was leaned against a wall at shoulder height. Two flaps were placed on top of the wall to restrict shoulder motion. The tests were considered valid if the subject did not move the shoulders or trunk away from the wall. Only the cervical spine with the head were allowed to move. In case of an error, the test was repeated. The described procedure was also used in a previous study [11].

The ET application was administered by a skilled operator immediately after T0. The same operator (T.P.) applied the ET for all subjects in the study group (SG). The ET application followed the Taping Elastico^®^ Method (ATS, Arezzo, Italy) and was applied symmetrically to both sides of the cervical area [34]. A strip of tape was cut into a “Y” shape, with the anchor applied to the skin at the level of the acromion and the two tails directed towards the base of the head. One tail followed the upper trapezius direction on the lateral portion of the neck, while the other was applied to the posterior portion (Figure 5). During the application, participants inclined their heads to the opposite side, stretching the skin. The ET was applied with zero tension, aiming to create convolutions when the head was in a neutral position. The ET application was bilateral and symmetrical.

After the ET application, participants in the SG waited for 20 min while wearing the application before proceeding to the next test session (T1), which was in line with previous study [29] and the manufacturer’s guidelines. The control group (CG) was observed at the same time between T0 and T1 without any intervention. Participants in the SG continued to wear the ET for three days during their daily activities. They were instructed to be cautious during dressing and washing to minimize the possibility of dislodging the ET. No instances of detachment were reported until T2. On the third day, one hour prior to the final test session (T2), the operator removed the ET.

### 2.4. Statistical Analysis

The normality of the data was assessed using the Shapiro–Wilk test. As the data followed a normal distribution, parametric tests were employed for the analysis. Differences at baseline were tested with a *t*-test for independent samples. A mixed ANOVA design (Time × Group) for repeated measures with Bonferroni correction was used to compare the post hoc effects (comparisons between T0 and T1, T1 and T2, and T0 and T2). Effect sizes (partial eta squared, $\eta_{p}^{2}$) were also calculated to facilitate the interpretation of the results, with values of 0.01, 0.06, and above 0.15 indicating small, medium, and large effect sizes, respectively [35,36]. The significance level was set at *p* = 0.05, and the statistical analysis was performed using SPSS (SPSS Inc., Chicago, IL, USA).

## 3. Results

The statistical analysis revealed no significant differences between SG and CG at T0 for any of the parameters measured. A Time × Group significant interaction was measured for all the ACROM directions, as well as for self-perceived pain, indicating that the changes in ROM and pain over time are not the same across the two groups (Table 1).

The CG did not exhibit any significant differences in any parameter at any time point. However, the SG demonstrated a significant increase in all ACROM values at T2 compared to T0 (Table 2). Additionally, self-perceived pain in the cervical area was significantly lower at T2 compared to T0. Significant differences were also observed between T0 and T1, as well as between T1 and T2. Table 2 provides a detailed overview of the results.

The ET application resulted in increased ACROM in all directions after three days of wearing. However, significant increases were specifically observed in left rotation and flexion, even after just 20 min of wearing. Additionally, when comparing the measurements taken at T1 and T2, all ACROM directions, except for the extension phase, showed significant increases (Figure 6).

Furthermore, the self-perceived cervical pain exhibited a significant decrease after both 20 min and three days of wearing the ET application. Notably, there was also a significant difference in pain levels between T1 and T2 (Figure 7).

Each group was analyzed by dividing them by gender, with the aim of understanding if the results of the entire sample could be influenced by gender differences. The behavior of male and female participants in both SG and CG was consistent with the behavior of the whole sample. SG showed significant differences between T0, T1, and T2 in both male and female individuals, while CG did not show any differences in any gender. Due to the paucity of female subjects in SG, the standard deviation was high, which affected the post hoc comparisons. Nevertheless, the ANOVA showed significant changes over time. Table 3 and Table 4 summarize the statistical analyses for male and female individuals, respectively.

## 4. Discussion

The primary objective of this study was to evaluate the immediate and short-term impacts of the application of an ET to the cervical area on self-perceived pain and the multiplanar motion of the cervical spine. The main novelty of this investigation lies in examining the short-term effects of a bilateral and symmetrical ET application on ACROM. This kind of approach can be considered a novelty because previous research did not use symmetrical applications. Additionally, investigating the effects of ET on pain is of great interest to healthcare professionals, given the chronic nature of cervical pain and its substantial societal costs [37,38,39]. In fact, cervical pain is recognized as one of the leading causes of global disability, placing it among the top five contributors [40].

Previous research conducted by Erdoğanoğlu et al. [30] with a similar study design demonstrated that wearing an ET application for 24 h resulted in a significant reduction in neck pain and improved ACROM. However, it is important to note that this previous study only included symptomatic individuals with cervical pain and lacked a control group; moreover, the observed effects were limited to a 24-h timeframe. In contrast, the present study also included symptomatic individuals but incorporated a control group and extended the measurement period by two days using a single ET application. This represents a significant advancement in the scientific evidence supporting the application of ET because it should be considered that the proposed methodology used in the current study is more similar to the everyday use of ET by individuals. Another study by Alahmari et al. [31] investigated the effects of ET application for more than three days, extending the period up to seven days, but it employed a different application technique and did not measure ACROM. Additionally, Ay et al. [32] demonstrated the effectiveness of five ET applications over a two-week period in terms of reducing neck pain and improving ACROM. While their study employed a similar ET application method to that used in our research, the ET shape, location, and unilateral application differed.

The findings of the present study align with previous literature (although different ET applications were used), supporting the immediate effects of ET on perceived pain and ACROM. Particularly noteworthy was the significant reduction in perceived pain after 20 min of ET application in the SG (7.5 ± 1.0 and 5.5 ± 1.4 at T0 and T1, respectively; *p* < 0.000). Moreover, in the same time span, there was a significant increase in ACROM for left rotation (64.8 ± 7.7° and 70.4 ± 12.5° at T0 and T1, respectively, *p* = 0.041) and flexion (55.0 ± 3.6° and 61.7 ± 12.8° at T0 and T1, respectively, *p* = 0.007). These immediate improvements in flexion and pain reduction confirm the positive effects of ET application. It is well recognized that individuals with cervical pain and disorders often experience limitations in flexion, making these findings particularly relevant [9,10,11,12,13].

From a practical and professional standpoint, the most notable finding of this study is undeniably the short-term effect observed after three days of using a single bilateral ET application. It is widely acknowledged that ET applications cannot be worn for extended periods due to factors such as personal hygiene practices, perspiration, and clothing changes. Consequently, it is common practice among healthcare professionals to remove and replace the ET application every three or four days [32,36]. The results of this study validate these procedural recommendations, demonstrating a substantial effect lasting for three days with a single ET application, without the need for removal and replacement.

For each direction of ACROM, a significant average improvement of 15% was observed between T0 and T2. Particularly noteworthy were the higher relative improvements observed in two specific directions: left rotation (64.8 ± 7.7° and 76.0 ± 11.1° at T0 and T2, respectively, *p* < 0.001) and extension (47.0 ± 12.9° and 55.1 ± 12.3° at T0 and T2, respectively, *p* < 0.001). These directions demonstrated a remarkable 17% increase in ACROM following three days of ET application. Furthermore, self-perceived pain exhibited a substantial average decrease of −81% between T0 and T2 (7.5 ± 1.0 and 1.4 ± 1.5, *p* < 0.001, respectively). It is relevant to highlight that the current results are gender independent, as both male and female participants showed significant improvements in ACROM and significant reductions in self-perceived pain. This aspect is highly relevant to public health because the absence of a gender effect enables professionals to apply the proposed method to a broad and diverse range of individuals without any gender-based restrictions.

One crucial aspect that deserves discussion in this paper is the different approach to the application of ET employed in this study compared to those of previous studies. While previous studies applied tension to the ET [30,31], with one exception that used a similar application method but a different placement and shape [32], in this study, the ET was applied without tension while stretching the skin during application (by inclining the head on the opposite side). This technique allowed the formation of skin convolutions when the head and neck were in a neutral position. It is very important to note that the two methods of application (with tension and without tension) differ significantly in terms of the pressure exerted on the skin. Although skin convolutions are thought to enhance local blood flow, the available data do not strongly support this claim [41,42,43]. While there are various studies of lower back pain that indicate no significant effects or differences between elastic taping applications with or without convolutions [44,45], there is a lack of similar data for the cervical area, except for the research conducted by Ay et al. [32]. They demonstrated that both convoluted and non-convoluted ET applications are effective in reducing neck pain and improving ACROM. Thus, the present study can be considered one of the first to use a bilateral and symmetrical ET application on the cervical area for three consecutive days, assessing its effects on self-perceived pain and ACROM. It is intriguing to hypothesize how two vastly different methods of ET (with and without tension) can yield similar results in terms of pain relief and cervical motion. A plausible explanation can be found in the existing literature, which suggests that the actual effect of ET can be attributed to both the direct contact itself and the pressure gradient generated between the taping and the surrounding skin area [46]. This idea is based on the findings of Pamuk et al. [47], who discuss the effects of ET on the underlying tissues, both at the immediate application site and in more distant regions. However, it should be noted that the latter explanation remains a hypothesis, and further research is required to elucidate this phenomenon. At present, the literature on ET continues to grapple with certain stigmas resulting from past inaccurate advertising that attributed false effects to taping. Therefore, studies like the present one are crucial for elucidating and enhancing the level of evidence regarding the use of ET for the treatment of musculoskeletal disorders.

### Limitations

Like any scientific study, even this study has certain limitations that need to be acknowledged. One notable limitation is the inability to conduct additional test sessions following a three-day washout period of ET. Consequently, the findings of this paper are specific to the immediate and short-term effects of ET, and information regarding the residual effects after the removal is currently unavailable. This aspect is crucial because individuals with musculoskeletal disorders are often concerned about the duration of treatment’s positive effects over time. Health professionals may find it valuable to comprehend the post-removal effects of ET and the duration of its benefits, enabling more precise intervention timing. Future research should aim to replicate our study protocol while incorporating a fourth and fifth test session to assess changes in self-perceived pain and multiplanar ACROM after ET removal. Since the restoration of muscle function is considered essential in the treatment of cervical spine disorders [15], it would be highly important to further investigate whether ET application can be regarded as beneficial for muscle function restoration.

The use of only one inertial sensor to detect cervical spine motion should be mentioned as a limitation. Although the inertial sensor showed results consistent with the 3D optoelectronic system and can be considered a valid tool for measuring cervical spine motion (see Appendix A), its use can introduce errors if the operator does not pay proper attention during the test. In fact, during this study, to reduce the margin of error, the shoulder position was fixed to avoid any compensatory motions. The authors strongly advise professionals to pay close attention to this aspect.

Another limitation of this study is the absence of a placebo group. Although previous results suggest the absence of a placebo effect for the cervical area [31,32] it would be interesting for future study designs to consider this critical aspect. For example, randomizing or having a crossover design would elicit some of this potential improvement.

In the end, although the current results can be considered gender independent because both male and female participants showed significant modifications induced via ET application, it is worth noting that the groups in this research are not gender-balanced. Specifically, the male-to-female ratio in this study and the control groups is 2.4 and 1.2, respectively. This imbalance is due to the participants lost during the preparation stage, as shown in Figure 1. It is possible that this aspect did not affect the results in any manner; however, it is fair to acknowledge its potential impacts. It would be interesting to replicate the experimental procedure with groups that have more balanced gender distributions.

## 5. Conclusions

In summary, the results of this study emphasize the effectiveness of wearing bilateral and symmetrical ET cervical applications for three days. To use ET is recommended for enhancing multiplanar ACROM and reducing self-reported cervical pain when needed, both in male and female subjects, especially among computer workers. This represents the primary novelty of this study. Specifically, applying ET without tension to create skin convolutions is a safe and cost-effective procedure for managing neck pain and improving the multiplanar motion of the cervical spine. These results can have immediate practical applications in the management of individuals’ musculoskeletal health, especially among computer workers. Ultimately, a three-day ET application can be recommended for managing neck motion restrictions and pain in adult individuals. Nevertheless, further research is required to extend the application of these findings to other domains, such as sports.

## Figures and Tables

**Figure 1 jfmk-08-00156-f001:**
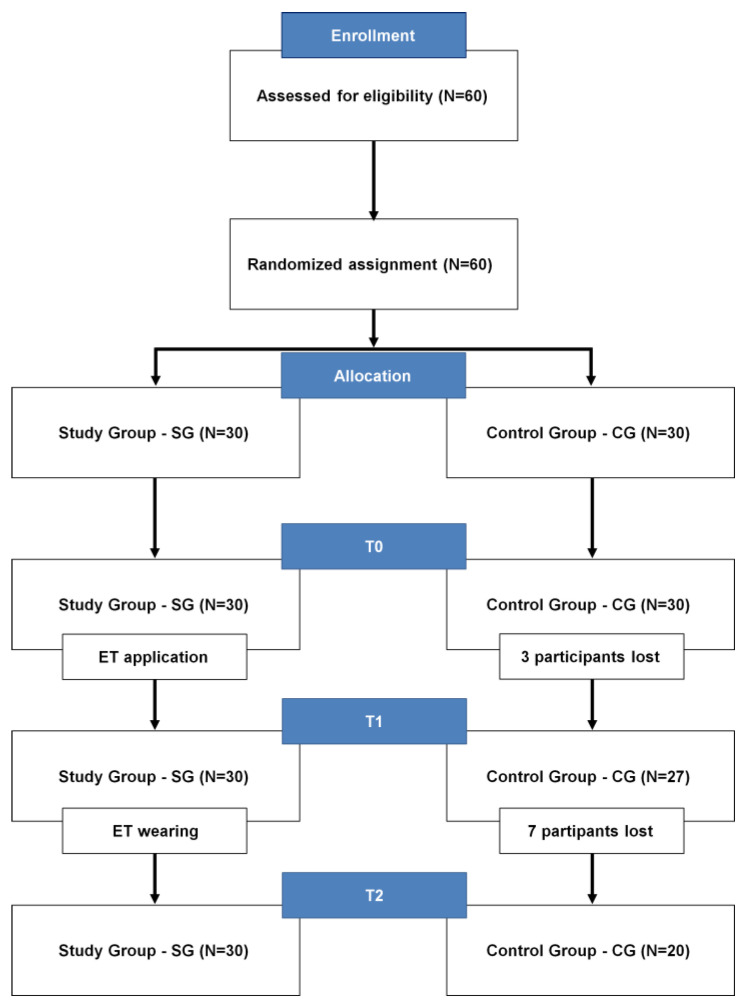
Participants’ distribution into groups. In particular, for CG, (1) three participants lost immediately after T0 expressed discomfort when performing the ACROM test and were afraid that it might cause more pain. As a result, they chose to withdraw from this study. (2) Seven subjects were lost between T1 and T2 because they opted to undergo some form of treatment, and, therefore, they were excluded from this analysis.

**Figure 2 jfmk-08-00156-f002:**
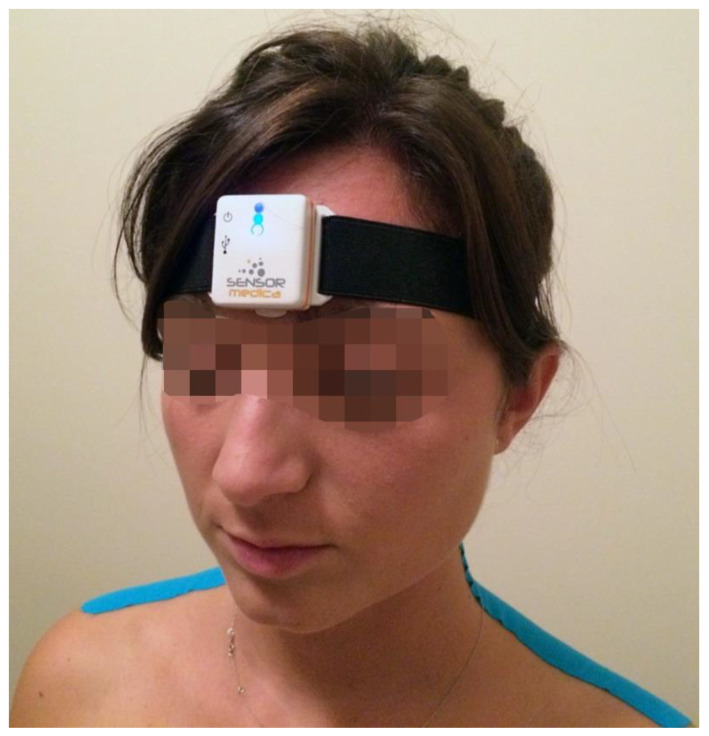
Placement of the inertial sensor on the forehead for measuring of the multiplanar ACROM.

**Figure 3 jfmk-08-00156-f003:**
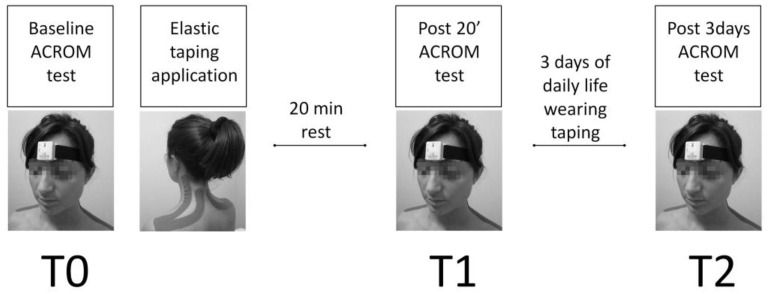
Graphical description of procedures and this study’s timeline.

**Figure 4 jfmk-08-00156-f004:**
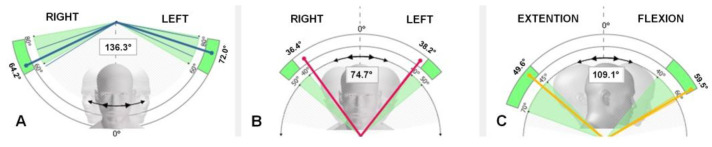
ACROM directions and anatomical planes tested using the inertial sensor: (**A**) right–left rotation on the transversal plane; (**B**) right–left lateral inclination on the frontal plane; (**C**) flexion–extension on the sagittal plane.

**Figure 5 jfmk-08-00156-f005:**
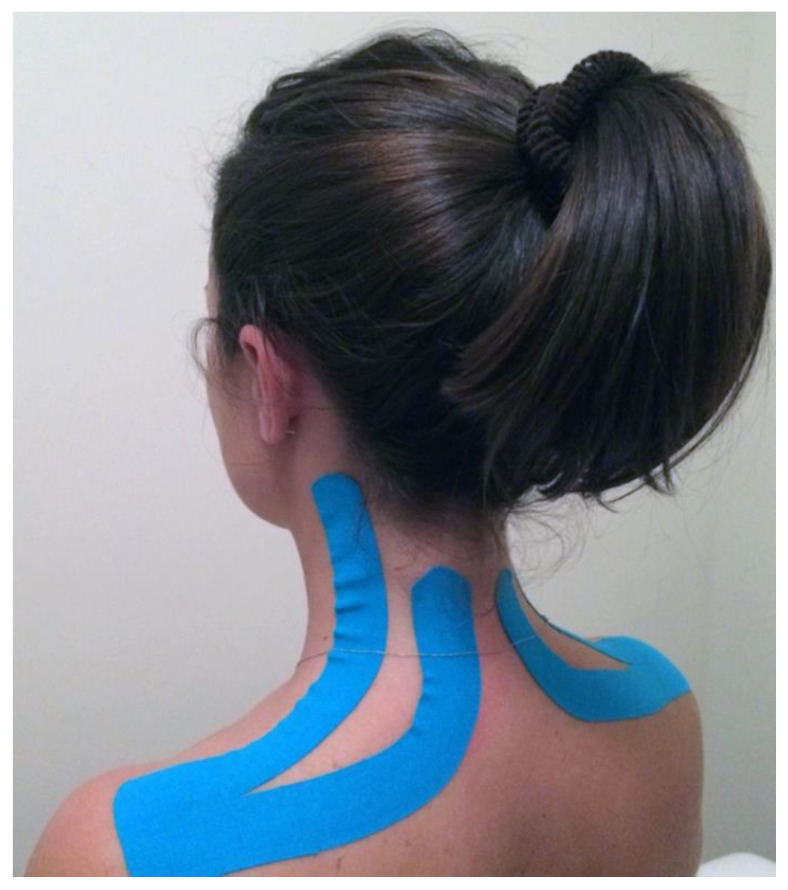
ET application following the Taping Elastico^®^ Method. It was possible to distinguish the convolutions of the two tails, typical of a zero-tension application with the aim of reducing compression of the underlying tissues.

**Figure 6 jfmk-08-00156-f006:**
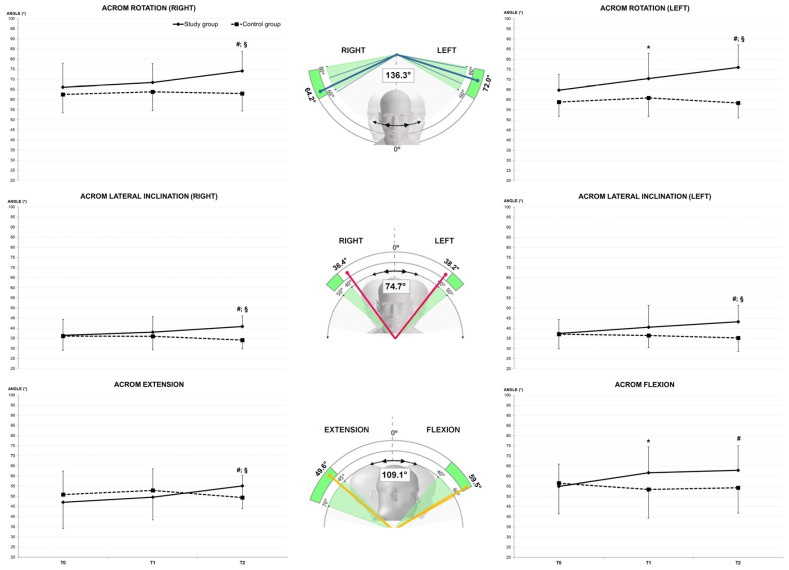
ACROM modifications over time in the six tested directions for SG (solid line) and CG (dotted line). “*”—significant difference between T0 and T1. “^#^”—significant difference between T0 and T2. “^§^”—significant difference between T1 and T2.

**Figure 7 jfmk-08-00156-f007:**
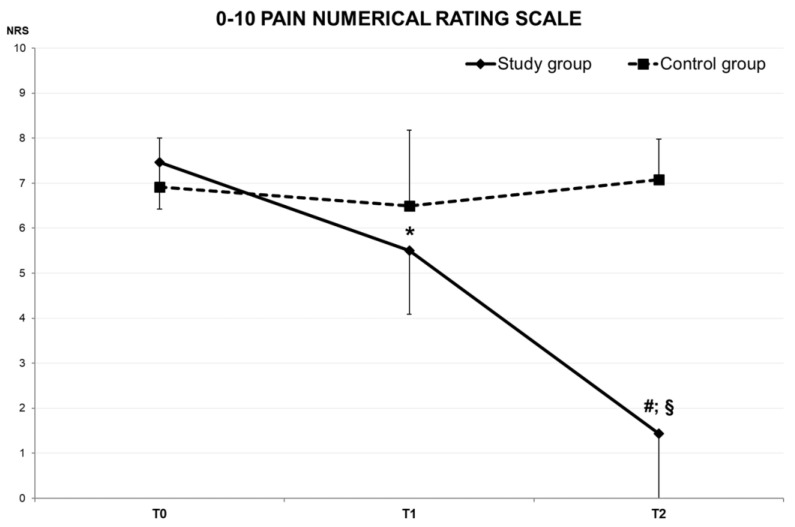
Numerical Rating Scale of self-perceived cervical pain modifications over time for SG (solid line) and CG (dotted line). “*”—Significant difference between T0 and T1. “^#^”—Significant difference between T0 and T2. “^§^”—Significant difference between T1 and T2.

**Table 1 jfmk-08-00156-t001:** Results of the mixed ANOVA Time × Group interaction.

Parameter	F	*p* Value
ACROM right rotation (°)	3.764	0.027
ACROM left rotation (°)	8.034	0.001
ACROM right lateral inclination (°)	10.400	0.000
ACROM left lateral inclination (°)	7.537	0.001
ACROM extension (°)	7.435	0.001
ACROM flexion (°)	5.708	0.005
Pain (A.U.)	113.111	0.000

Note: ACROM—active cervical range of motion.

**Table 2 jfmk-08-00156-t002:** Results of the experiment for SG and CG.

	Parameter	T0	T1	T2	$\eta_{p}^{2}$	*p* Value
Study Group (SG)	ACROM right rotation (°)	66.0 ± 11.9	68.4 ± 9.3	74.2 ± 9.6 ^#§^	0.432	0.000
	ACROM left rotation (°)	64.8 ± 7.7	70.4 ± 12.5 *	76.0 ± 11.1 ^#§^	0.599	0.000
	ACROM right lateral inclination (°)	36.5 ± 7.9	38.0 ± 7.9	40.9 ± 5.2 ^#§^	0.452	0.000
	ACROM left lateral inclination (°)	37.5 ± 6.9	40.6 ± 10.8	43.2 ± 8.0 ^#§^	0.554	0.000
	ACROM extension (°)	47.0 ± 12.9	49.6 ± 11.3	55.1 ± 11.3 ^#§^	0.360	0.000
	ACROM flexion (°)	55.0 ± 3.6	61.7 ± 12.8 *	62.9 ± 12.0 ^#^	0.288	0.002
	Pain (A.U.)	7.5 ± 1.0	5.5 ± 1.4 *	1.4 ± 1.5 ^#§^	0.945	0.000
Control Group (CG)	ACROM right rotation (°)	62.5 ± 9.1	63.8 ± 9.3	63.0 ± 8.7	0.017	0.569
	ACROM left rotation (°)	58.8 ± 7.0	60.9 ± 9.2	58.4 ± 7.5	0.012	0.630
	ACROM right lateral inclination (°)	36.0 ± 7.0	36.0 ± 6.7	34.1 ± 4.2	0.094	0.175
	ACROM left lateral inclination (°)	37.1 ± 7.1	36.4 ± 6.0	35.2 ± 6.6	0.009	0.675
	ACROM extension (°)	50.9 ± 11.6	52.9 ± 10.7	49.3 ± 10.8	0.049	0.337
	ACROM flexion (°)	56.5 ± 15.2	53.4 ± 14.1	54.3 ± 12.5	0.081	0.211
	Pain (a.u.)	6.9 ± 1.1	6.5 ± 1.7	7.1 ± 0.9	0.056	0.438

Note: ACROM—active cervical range of motion. T0—baseline test session. T1—test session after 20′ of wearing ET application. T2—test session after three days of wearing ET application. “*”—significant difference between T0 and T1. “^#^”—significant difference between T0 and T2. “^§^”—significant difference between T1 and T2.

**Table 3 jfmk-08-00156-t003:** Results of the experiment for male participants in SG and CG.

	Parameter	T0	T1	T2	$\eta_{p}^{2}$	*p* Value
Study Group (SG)	ACROM right rotation (°)	65.9 ± 11.0	68.2 ± 7.6	74.3 ± 7.4 ^#§^	0.415	0.000
	ACROM left rotation (°)	64.7 ± 8.1	70.6 ± 9.9 *	75.9 ± 7.9 ^#§^	0.680	0.001
	ACROM right lateral inclination (°)	35.9 ± 7.4	37.5 ± 7.4	40.7 ± 4.9 ^#§^	0.536	0.000
	ACROM left lateral inclination (°)	36.0 ± 6.0	38.9 ± 7.6	42.5 ± 6.9 ^#§^	0.626	0.000
	ACROM extension (°)	47.3 ± 13.1	51.7 ± 8.0	55.1 ± 9.4 ^#^	0.327	0.004
	ACROM flexion (°)	55.6 ± 10.8	62.5 ± 11.3 *	63.2 ± 10.6	0.241	0.017
	Pain (A.U.)	7.5 ± 1.1	5.8 ± 1.3 *	1.5 ± 1.6 ^#§^	0.938	0.000
Control Group (CG)	ACROM right rotation (°)	64.3 ± 9.0	64.4 ± 6.9	64.4 ± 6.7	0.002	0.892
	ACROM left rotation (°)	58.9 ± 7.2	62.7 ± 8.5	59.0 ± 6.9	0.000	0.948
	ACROM right lateral inclination (°)	38.2 ± 7.1	38.1 ± 5.7	35.6 ± 3.1	0.279	0.077
	ACROM left lateral inclination (°)	38.4 ± 7.4	38.5 ± 5.5	36.8 ± 6.9	0.141	0.229
	ACROM extension (°)	47.6 ± 13.6	50.0 ± 12.3	45.1 ± 11.2	0.079	0.376
	ACROM flexion (°)	57.8 ± 13.9	53.5 ± 13.0	56.0 ± 10.4	0.034	0.568
	Pain (a.u.)	4.3 ± 3.5	3.8 ± 3.3	4.6 ± 3.7	0.364	0.058

Note: ACROM—active cervical range of motion. T0—baseline test session. T1—test session after 20′ of wearing ET application. T2—test session after three days of wearing ET application. “*”—significant difference between T0 and T1. “^#^”—significant difference between T0 and T2. “^§^”—significant difference between T1 and T2.

**Table 4 jfmk-08-00156-t004:** Results of the experiment for female participants in SG and CG.

	Parameter	T0	T1	T2	$\eta_{p}^{2}$	*p* Value
Study Group (SG)	ACROM right rotation (°)	66.5 ± 15.2	69.0 ± 13.6	73.7 ± 14.9	0.523	0.028
	ACROM left rotation (°)	65.1 ± 6.7	69.9 ± 18.8	76.3 ± 18.0	0.451	0.047
	ACROM right lateral inclination (°)	38.1 ± 9.5	39.4 ± 9.3	41.4 ± 6.3	0.256	0.165
	ACROM left lateral inclination (°)	41.6 ± 8.0	45.1 ± 16.7	45.3 ± 10.8	0.355	0.091
	ACROM extension (°)	46.2 ± 13.0	43.8 ± 16.8	55.1 ± 16.3 ^§^	0.466	0.043
	ACROM flexion (°)	53.2± 12.2	59.5 ± 16.9	62.2 ± 16.2	0.484	0.037
	Pain (A.U.)	7.3 ± 0.7	4.8 ± 1.4 *	1.1 ± 1.5 ^#§^	0.965	0.000
Control Group (CG)	ACROM right rotation (°)	60.4 ± 10.4	63.1 ± 12.0	61.1 ± 10.9	0.072	0.453
	ACROM left rotation (°)	58.7 ± 7.2	58.7 ± 10.0	57.7 ± 8.4	0.078	0.434
	ACROM right lateral inclination (°)	33.4 ± 6.2	33.3 ± 7.1	32.3 ± 4.9	0.129	0.308
	ACROM left lateral inclination (°)	35.5 ± 6.9	33.9 ± 5.9	33.2 ± 6.0	0.318	0.090
	ACROM extension (°)	54.8 ± 7.5	56.3 ± 7.7	54.4 ± 8.3	0.011	0.770
	ACROM flexion (°)	55.0 ± 17.3	53.3 ± 16.3	51.9 ± 15.0	0.185	0.215
	Pain (a.u.)	4.0 ± 3.9	4.1 ± 3.9	3.9 ± 3.5	0.053	0.594

Note: ACROM—active cervical range of motion. T0—baseline test session. T1—test session after 20′ of wearing ET application. T2—test session after three days of wearing ET application. “*”—significant difference between T0 and T1. “^#^”—significant difference between T0 and T2. “^§^”—significant difference between T1 and T2.

## Data Availability

The data that support the findings of this study are available from the corresponding author upon reasonable request.

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
