# Peer review of "Elastic Taping Application on the Neck: Immediate and Short-Term Impacts on Pain and Mobility of Cervical Spine"

_jfmk, 2023, doi:10.3390/jfmk8040156_

Round 1

Reviewer 1 Report (New Reviewer)

Comments and Suggestions for Authors

Elastic Tape Application on the Neck: Immediate and Short-Term Results on Pain and Mobility of Cervical Spine

The manuscript topic is important and socially significant, supported with experiment. The work is valuable for its practical focus on human health. Everything related to health is valuable and socially significant.

1. The Reference 44 has not an publication year.

2. The Abstract and Conclusions must be improved.

Improvement. In the abstract and conclusions must be underline clearly the new results and conclusions presented from the authors which differ from those obtained till now. In the manuscript text there are these first time benefits. In the Abstract must be included the main conclusions and the authors must underline the own approach contributions and the basic benefits from presented results in practice. The unwritten rule is that most readers only look at these paragraphs – abstract and conclusions.

I hope that the proposed corrections will increase the quality of the manuscript and possibly its citability.

Author Response

Dear Reviewer 1,

We would like to thank you for the time allowed to this review process. As a result, we are submitting the revised version for a possible publication in this respectable Journal. Below, you can find our responses; each comment is followed by its respective reply. We made changes in the manuscript in order to address suggestions and make it clearer for the readers. Our responses in the manuscript appear using the track change instrument. We very much appreciate your comments on the document, which have helped us to improve its quality.

All authors have made sufficient contributions, responded to your comments and have approved the submitted manuscript.

Best regards,

The Authors

Legend:

R1(Reviewer 1)

A (Authors)

1) R1:

The manuscript topic is important and socially significant, supported with experiment. The work is valuable for its practical focus on human health. Everything related to health is valuable and socially significant.

A: Thanks for the comment, we are glad to see that our work has been appreciated.

2) R1:

The Reference 44 has not an publication year.  

A: Done. Please note that reference 44 is now reference 42 because the editor suggested us to review the references list and two references have been removed.

3) R1:

The Abstract and Conclusions must be improved.

Improvement. In the abstract and conclusions must be underline clearly the new results and conclusions presented from the authors which differ from those obtained till now. In the manuscript text there are these first time benefits. In the Abstract must be included the main conclusions and the authors must underline the own approach contributions and the basic benefits from presented results in practice. The unwritten rule is that most readers only look at these paragraphs – abstract and conclusions. 

A:  We are grateful for the comment. It gave us the possibility to improve the clarity of the conclusions. We briefly modified the abstract (lines 35-39) and the conclusions as well (lines 382-394).

4) R1:

I hope that the proposed corrections will increase the quality of the manuscript and possibly its citability. 

A: We are sure that your proposals will enhance the quality of our paper.

Reviewer 2 Report (New Reviewer)

Comments and Suggestions for Authors

Dear authors.

I think the article is interesting and adds evidence to the present investigation.

I wish I had found a sample size calculation in the methods section

However, it seems necessary to reflect on the following question.

The groups are not homogeneous in gender and therefore this is a limitation that should be discussed in the limitations section.

Of course they can do a statistical analysis before and after and with it comment how they do in lines 310-315 say that in both sexes after the application of the elastic bandage the ROM improves.

But they should discuss limitations that may influence the result that the control group has a higher percentage of women.

Author Response

Dear Reviewer 2,

We would like to thank you for the time allowed to this review process. As a result, we are submitting the revised version for a possible publication in this respectable Journal. Below, you can find our responses; each comment is followed by its respective reply. We made changes in the manuscript in order to address suggestions and make it clearer for the readers. Our responses in the manuscript appear using the track change instrument. We very much appreciate your comments on the document, which have helped us to improve its quality.

All authors have made sufficient contributions, responded to your comments and have approved the submitted manuscript.

Best regards,

The Authors

Legend:

R2 (Reviewer 2)

A (Authors)

1) R2:

I think the article is interesting and adds evidence to the present investigation.

A: Thanks for the comment, we are glad to see that our work has been appreciated.

2) R2:

The groups are not homogeneous in gender and therefore this is a limitation that should be discussed in the limitations section.

Of course they can do a statistical analysis before and after and with it comment how they do in lines 310-315 say that in both sexes after the application of the elastic bandage the ROM improves.

But they should discuss limitations that may influence the result that the control group has a higher percentage of women.  

A: Thanks for the comment. We add a paragraph in “Limitations” section, lines 371-378.

This manuscript is a resubmission of an earlier submission. The following is a list of the peer review reports and author responses from that submission.

Round 1

Reviewer 1 Report

Comments and Suggestions for Authors

General Comments

1.     It is highly likely that there is a significant placebo effect of this taping technique. There is a relatively large body of research highlighting the impact of similar tapes is negligible and highly influenced by the placebo effect. There needs to be at least a passing mention of this in the limitations. Recognizing this is critical and should be included in future study designs. For example, randomizing or having a cross over design would elicit some of this possibility.

Abstract

1.     Lines 22-24: There is a significant sample size difference between men and women in your control group. Were there any differences between the sexes in either cohort? Were there significant differences by sex across the two cohorts?

Introduction

1.     Line 45: These are atypical terms for the types of motions you are describing. More typical would be flexion/extension in the sagittal plane. Strongly consider modifying these to better match well-established biomechanics terminology. Change throughout the document.

2.     Lines 49-60: These two paragraphs should be combined.

3.     Line 66-67: This is highly debatable. Use of an IMU for the any spinal segment is very challenging because it becomes an estimated ROM of head relative to the torso (typically). As such, you don’t know where that change in motion is occurring (i.e. at what bony segment). Furthermore, it is unclear in this study if you’re really measuring cervical or head motion because you are only looking at 1 IMU. That means you only really know the orientation of the bony segment of interest and not the relative motion of that bony segment to the corresponding distal or proximal segment.

4.     Lines 84-86: I am not sure I agree that this is a reasonable statement given the sentiments before it. You need to have some literature sources to back up why this is a reasonable hypothesis.

Methods

1.     Were any interventions received prior to enrolling? Most people with neck pain will seek treatment to modify their motion/pain (e.g. chiropractic care, injections, etc.). This would heavily influence your result.

2.     How was chronic defined herein?

3.     It is a bit suspicious that all 10 subjects who withdrew from the study were in the control group. The odds of that happening and having nobody drop from the treatment group is nearly 0. Explain why this happened.

4.     Lines 103-105: It is statistically inappropriate to run a power analysis after your data are collected. A power analysis is conducted in order to define the number of subjects you need to enroll in advance of collecting that data.

5.     Line 114-132: This is the most limiting aspect of your method. You are utilizing 1 Imu attached to the forehead. That gives you some information about the orientation of the head. But without a second sensor somewhere down the kinematic chain, you can’t definitively say what the relative motion is of the head relative to any other segment. For example, if a subject slouched just a little bit but kept their head in the same posture, the IMU would quantify the same “neck motion” but in reality have significantly more cervical spine flexion. Given your results are statistically significant over very small ROM (e.g. 3-5 degrees in some cases), I am unconvinced that what you found is truly what is happening. The only way to ensure this motion was actually all occurring in the cervical spine was to physically restrain the torso and make sure it was in the exact same place and orientation for every movement. This is highly unlikely.

6.     Lines 163-172: Why did you choose inferior to superior taping approach? Why not superior to inferior? Would you expect to see a difference? You discuss this a bit in the discussion about two taping approaches achieving the same result. If that’s the case, does this mean the tape is arbitrary and really this is the placebo effect interaction again?

7.     Line 177: Why was 20 minutes chosen?

8.     Line 190-192: Why were these specific effect sizes chose to differentiate small, medium, and large ES?

Results

1.     Table 2: It is unclear how you are able to say that two specific time points were different than one another if you only conducted an ANOVA. If you’re running an ANOVA, that only tells that there is an impact of time but can’t say specifically when that difference occurred. Can you clarify this in your statistical methods section?

Author Response

Dear Reviewer 1,

We would like to thank you for the time allowed to this review process. As a result, we are submitting the revised version for a possible publication in this respectable Journal. Below, you can find our responses; each comment is followed by its respective reply. We made changes in the manuscript in order to address suggestions and make it clearer for the readers. Our responses in the manuscript appear using the track change instrument. We very much appreciate your comments on the document, which have helped us to improve its quality.

All authors have made sufficient contributions, responded to your comments and have approved the submitted manuscript.

Best regards,

The Authors

Legend:

R1(Reviewer 1)

A (Authors)

1) R1:

It is highly likely that there is a significant placebo effect of this taping technique. There is a relatively large body of research highlighting the impact of similar tapes is negligible and highly influenced by the placebo effect. There needs to be at least a passing mention of this in the limitations. Recognizing this is critical and should be included in future study designs. For example, randomizing or having a cross over design would elicit some of this possibility.

A: Thanks for the comment. We added this important point of view in lines 349-353.

2) R1:

Lines 22-24: There is a significant sample size difference between men and women in your control group. Were there any differences between the sexes in either cohort? Were there significant differences by sex across the two cohorts?  

A: We add more statistical informations about the behavior for each gender in lines 243-260, 310-312 and 359. This crucial point was really helpful to improve the results section in this manuscript.

3) R1:

Line 45: These are atypical terms for the types of motions you are describing. More typical would be flexion/extension in the sagittal plane. Strongly consider modifying these to better match well-established biomechanics terminology. Change throughout the document. 

A: Thanks for this comment, we agree with your opinion, therefore we modified the terms throughout the main document. Originally, we did that choice because it is common to find a different use of the term’s flexion/extension in the sagittal plane. Sometimes flexion is considered the forward direction, sometimes the backward direction (because the lordosis of the cervical spine). To avoid any kind of further misunderstanding we use flexion (meaning forward head direction) and extension (meaning backward head direction). This point is specifically descripted in line 46.

4) R1:

Lines 49-60: These two paragraphs should be combined. 

A: Done, thanks for the suggestion.

5) R1:

Lines 84-86: I am not sure I agree that this is a reasonable statement given the sentiments before it. You need to have some literature sources to back up why this is a reasonable hypothesis. 

A: Thanks for the comment. We can try to explain better. Starting from current line 80 we wrote “Based on current knowledge, the effectiveness of ET applications in reducing neck pain and improving ACROM has been demonstrated [30-32].” The references 30-32 affirm that the improvement of ACROM happens; this is reason why we wrote “It is reasonable” in line 84. However, we modified the text in line 84.

6) R1:

Were any interventions received prior to enrolling? Most people with neck pain will seek treatment to modify their motion/pain (e.g. chiropractic care, injections, etc.). This would heavily influence your result. 

A: Thanks for the comment, it is a very interesting point to discuss. In fact, we enrolled only subjects who did not undergo to any kind of treatment. We added this information in the lines 96-97.

7) R1:

How was chronic defined herein? 

A: We can understand that the way we wrote cannot be clear. We used the term chronic in a wrong way aiming to reinforce the recurrence of the self-perceived musculoskeletal cervical pain. We deleted “chronic” adding the rate of recurrence in line 95-96.

8) R1:

It is a bit suspicious that all 10 subjects who withdrew from the study were in the control group. The odds of that happening and having nobody drop from the treatment group is nearly 0. Explain why this happened.

A: We understand the reviewer's suspicion, but this can be easily explained. The participants lost immediately after T0 expressed discomfort in performing the ACROM test and were afraid that it might cause more pain. As a result, they choose to withdraw from our study. On the other hand, the seven subjects lost between T1 and T2 opted to undergo some form of treatment (see point 7), and therefore, they were excluded from the analysis. We have provided an explanation of these participants' history in the footnote of Figure 1, line 116-119.

9) R1:

Lines 103-105: It is statistically inappropriate to run a power analysis after your data are collected. A power analysis is conducted in order to define the number of subjects you need to enroll in advance of collecting that data.

A: Thanks for your comment, we agree with the reviewer. Anyway, we made the power analysis considering 30  participants for SG an CG but after we lose the CG participants (see point 9) we run again the power analysis to show how the loss of participants did not affect the procedures.

10) R1: NOTE: we merge two reviewer’s comment in this part because the topic is very similar and we preferred to give a single but complete answer to both comments.

Line 66-67: This is highly debatable. Use of an IMU for the any spinal segment is very challenging because it becomes an estimated ROM of head relative to the torso (typically). As such, you don’t know where that change in motion is occurring (i.e. at what bony segment). Furthermore, it is unclear in this study if you’re really measuring cervical or head motion because you are only looking at 1 IMU. That means you only really know the orientation of the bony segment of interest and not the relative motion of that bony segment to the corresponding distal or proximal segment. 

Line 114-132: This is the most limiting aspect of your method. You are utilizing 1 Imu attached to the forehead. That gives you some information about the orientation of the head. But without a second sensor somewhere down the kinematic chain, you can’t definitively say what the relative motion is of the head relative to any other segment. For example, if a subject slouched just a little bit but kept their head in the same posture, the IMU would quantify the same “neck motion” but in reality have significantly more cervical spine flexion. Given your results are statistically significant over very small ROM (e.g. 3-5 degrees in some cases), I am unconvinced that what you found is truly what is happening. The only way to ensure this motion was actually all occurring in the cervical spine was to physically restrain the torso and make sure it was in the exact same place and orientation for every movement. This is highly unlikely.

A: We combined these two comments because the topic is very similar and we give one only answer for this aspect. We well know what you underline therefore we performed the test locking the trunk and the shoulders. We took for granted this aspect because we well know what you told us therefore it is our fault because we did not explained in details the specific procedure we used to perform the test. We add this information in lines 165-170.

11) R1:

Lines 163-172: Why did you choose inferior to superior taping approach? Why not superior to inferior? Would you expect to see a difference? You discuss this a bit in the discussion about two taping approaches achieving the same result. If that’s the case, does this mean the tape is arbitrary and really this is the placebo effect interaction again?

A: Thank you for your comment; it gives us with the opportunity to provide further clarification. The decision to start the taping from the acromion and direct the two tails towards the base of the head (an inferior to superior approach) was mainly influenced by the specific application method we used. The "Y" tape we utilized could not be properly applied with a superior to inferior approach. Additionally, the choice of the inferior/superior approach aligned with the Taping Elastico method (reference 36), with the AY et al. paper (reference 32 - which, in turn, followed the Kenzo Kase method). It is essential to highlight that the tape was applied on stretched skin (line 178) without tension to create convolutions when the skin was in a non-stretched position. For this reason, when the tape is applied without tension on the stretched skin, the application direction becomes less critical: some studies suggest an inferior/superior approach (https://www.sciencedirect.com/science/article/pii/S1836955314000368?via%3Dihub, https://www.sciencedirect.com/science/article/pii/S2255502116300062?via%3Dihub#bib0115), while others propose a superior/inferior approach (https://www.ncbi.nlm.nih.gov/pmc/articles/PMC5776879/) with similar results. Conversely, the superior/inferior approach appears to be more effective when tension is applied to the tape (https://www.ncbi.nlm.nih.gov/pmc/articles/PMC5300808/).

12) R1:

Line 177: Why was 20 minutes chosen?

A: Thank you for your comment; it provides us with the opportunity to offer further details regarding this choice. We allowed a 20-minute interval between the application of the tape and the acute ACROM test for two specific reasons: 1) to replicate a similar time frame utilized in a previous study (reference 29); and 2) because the manufacturer of the taping we used recommends a 20-minute period to ensure optimal adherence with the skin. We have now included this information in lines 191-192 of the manuscript.

13) R1:

Line 190-192: Why were these specific effect sizes chose to differentiate small, medium, and large ES?

A: We added the references in line 206.

14) R1:

Table 2: It is unclear how you are able to say that two specific time points were different than one another if you only conducted an ANOVA. If you’re running an ANOVA, that only tells that there is an impact of time but can’t say specifically when that difference occurred. Can you clarify this in your statistical methods section?

A: The statistical approach used a mixed ANOVA design (Time x Group) for repeated measures with Bonferroni correction to compare the post-hoc effects. In any case, a better explanation is given in line 203.

Reviewer 2 Report

Comments and Suggestions for Authors

Review of the manuscript:

“Immediate and Short-Term Effects on Pain and Multiplanar Active Cervical Range of Motion of an Elastic Taping Application on the Neck”

The aim of the reviewed manuscript was to measure the effects on 3D active cervical range of motion and perceived pain of an elastic taping application in the cervical area. To achieve the goal, the Authors utilized a short-term longitudinal small cohort design with repeated measures. The selection of the research group and method was adequate to the research objective. The material and methods are described in detail and thoroughly. I only think that giving the average values of body mass and height of the surveyed people without division into sex is a mistake. I suggest, to presenet these values separately for women and men. The method of carrying out the measurements and putting on the elastic tape application is clear and allows the research to be repeated by other researchers. Statistical analysis was carried out correctly using appropriate tests. The results were presented graphically in a legible way. The authors extensively referred the results of their research to the available literature. They also provided the application of the obtained results.

Author Response

Dear Reviewer 2,

We would like to thank you for the time allowed to this review process. As a result, we are submitting the revised version for a possible publication in this respectable Journal. Below, you can find our responses; each comment is followed by its respective reply. We made changes in the manuscript in order to address suggestions and make it clearer for the readers. Our responses in the manuscript appear using the track change instrument. We very much appreciate your comments on the document, which have helped us to improve its quality.

All authors have made sufficient contributions, responded to your comments and have approved the submitted manuscript.

Best regards,

The Authors

Legend:

R2 (Reviewer 2)

A (Authors)

1) R2:

The aim of the reviewed manuscript was to measure the effects on 3D active cervical range of motion and perceived pain of an elastic taping application in the cervical area. To achieve the goal, the Authors utilized a short-term longitudinal small cohort design with repeated measures. The selection of the research group and method was adequate to the research objective. The material and methods are described in detail and thoroughly.

A: Thanks for the comment, we are glad to see that our work has been appreciated.

2) R2:

I only think that giving the average values of body mass and height of the surveyed people without division into sex is a mistake. I suggest, to presenet these values separately for women and men.  

A: Thanks for your comment, we modified the way to present the data dividing for gender the entire group. We modified this information in “Methods”, maintaining the previous modality of description in the abstract, avoiding overloading excessively the abstract. We added this information in lines 103-107.

3) R2:

The method of carrying out the measurements and putting on the elastic tape application is clear and allows the research to be repeated by other researchers. Statistical analysis was carried out correctly using appropriate tests. The results were presented graphically in a legible way. The authors extensively referred the results of their research to the available literature. They also provided the application of the obtained results. 

A: Thanks for the comment, we are glad to see that our work has been appreciated.